Enhancing Alzheimer’s disease classification through split federated learning and GANs for imbalanced datasets

Narayanee Nimeshika G
D Subitha subitha.d@vit.ac.in
School of Computer Science and Engineering, Vellore Institute of Technology , Chennai , Tamil Nadu , India
Balas Valentina Emilia
Electronic publication date: 2024 Nov 29
Publication date: 2024
Volume: 10
Electronic Location ID: e2459
Received 2024 Apr 11; Accepted 2024 Oct 7
Copyright: ©2024 Narayanee Nimeshika and Subitha
Copyright year: 2024
Copyright holder: Narayanee Nimeshika and Subitha
License: This is an open access article distributed under the terms of the Creative Commons Attribution License, which permits unrestricted use, distribution, reproduction and adaptation in any medium and for any purpose provided that it is properly attributed. For attribution, the original author(s), title, publication source (PeerJ Computer Science) and either DOI or URL of the article must be cited.
License URL: https://creativecommons.org/licenses/by/4.0/

Keywords: Decentralized, Imbalanced, Data privacy, Split federated learning, Conditional generative adversarial networks, Alzheimer

Funding: The authors received no funding for this work.

==============================
In the rapidly evolving healthcare sector, using advanced technologies to improve medical classification systems has become crucial for enhancing patient care, diagnosis, and treatment planning. There are two main challenges faced in this domain (i) imbalanced distribution of medical data, leading to biased model performance and (ii) the need to preserve patient privacy and comply with data protection regulations. The primary goal of this project is to develop a medical classification model for Alzheimer’s disease detection that can effectively learn from decentralized and imbalanced datasets without compromising on data privacy. The proposed system aims to address these challenges by employing an approach that combines split federated learning (SFL) with conditional generative adversarial networks (cGANs) to enhance medical classification models. SFL enables efficient set of distributed agents that collaboratively train learning models without sharing their data, thus improving data privacy and the integration of conditional GANs aims to improve the model’s ability to generalize across imbalanced classes by generating realistic synthetic samples for minority classes. The proposed system provided an accuracy of approximately 83.54 percentage for the Alzheimer’s disease classification dataset.

Introduction

Alzheimer’s disease is a progressive neurodegenerative disorder characterized by cognitive decline, memory loss, and impaired daily functioning  (Veitch et al., 2022). As the most common form of dementia, it affects millions of individuals worldwide, posing significant challenges for both patients and their families. Early detection of Alzheimer’s disease is crucial for timely intervention, as it allows for better management of symptoms and potential treatment strategies. Various medical imaging techniques, such as magnetic resonance imaging (MRI), have been instrumental in aiding the diagnosis of Alzheimer’s disease by revealing structural changes in the brain associated with the condition. In the wide landscape of healthcare, the integration of advanced technologies plays a major role in refining medical classification systems, thereby contributing to the improvement of patient care, accurate diagnosis, and effective treatment planning. In Mammen (2021), federated learning (FL) is defined as a machine learning technique that allows model training across decentralized devices or servers holding local data samples, without exchanging raw data. This collaborative approach enables the development of machine learning models while preserving the privacy of individual datasets. Instead of aggregating data in a centralized repository, federated learning distributes the model training process across multiple devices, with each device updating the model based on its local data. The main advantage of FL is parallel processing which takes place in the client side.

Poirot et al. (2019) propose a split learning (SL) based approach, a type of distributed learning, and apply it for the first time in the medical field. In SL, the model undergoes a division into two segments, with one segment situated on the client side and the other on the server side. This partition occurs at a specific layer which is known as the cut layer. The central server holds the initial model, and only updates or gradients are exchanged during training. This architecture allows collaborative learning across multiple participants without sharing raw data, making it particularly suitable for applications in healthcare and other privacy-sensitive domains.

Split federated learning (SFL) (Hafi et al., 2024) tries to combine the strengths of both FL and SL. Utilizing the parallel processing from FL and the model splitting concept from SL which enhances the accuracy and other evaluation metrics. Hafi et al. (2024) thoroughly investigates the implementation of collaborative learning, particularly SFL, in 6G wireless networks. The transition to small and distributed data encourages the adoption of collaborative machine learning (ML)/deep learning (DL) techniques, with SFL emerging as a promising approach. The research underscores the potential benefits of leveraging SplitFed’s distributed learning capabilities in the context of 6G networks. The challenges involved in SFL is splitting strategy, imbalanced data, data labeling, aggregation techniques, wireless channel constraints.

Data imbalance occurs when certain classes or categories in a dataset are significantly underrepresented compared to others, leading to potential biases in the model’s performance. This imbalance can adversely impact the performance of machine learning models, as they may become biased towards the majority class. Addressing data imbalance is crucial for ensuring that models generalize well across all classes, preventing skewed predictions and enhancing overall predictive accuracy. Generative adversarial networks (GANs) play a vital role in overcoming data imbalance by generating synthetic samples for minority classes. In the context of imbalanced datasets, GANs can produce realistic and diverse examples of underrepresented categories, thereby augmenting the training data. This augmentation helps balance class distribution, enhancing the model’s ability to learn and generalize across all classes, ultimately improving overall predictive performance.

The methodology adopted in this study involves the integration of SFL and GANs. SFL, a technique, facilitates the training of machine learning models through a collaborative effort of distributed agents without the need to share raw data. In a hospital with multiple branches, a central server can collect data from various branches (clients). However, a challenge arises when certain branches lack sufficient data for specific classes, leading to data imbalance. To address this issue, the hospital can employ cGAN to generate synthetic data and mitigate the imbalance. SFL is used because medical data is often sensitive and subject to strict privacy regulations. Also, in SFL each client does not require extensive more computing resources because the model architecture is split. SFL’s ability to split the network architecture between clients and the server enhances privacy by keeping certain aspects of the machine learning model localized on client devices. This ensures that patient data remains secure and confidential.

Related Work

The methodology of Kafshgari, Bajić & Saeedi (2023) involves transferring feature values, gradient updates, and model updates across communication channels in SplitFed learning. The article focuses on the effects of noise in the communication channels on the learning process and the quality of the final model, but it does not explore other potential limitations or challenges that may arise in the context of embryo image segmentation.

The research presented in article Afzal et al. (2019) focuses on the application of transfer learning (TL) for efficient classification of Alzheimer’s disease (AD) stages, particularly addressing the challenges of classifying patients into distinct stages of No Dementia, Very Mild Dementia, Mild Dementia, and Moderate Dementia. The proposed methodology involves a transfer learning-based technique, leveraging pre-trained AlexNet models, and focuses on two distinct perspectives: the main view of the brain and the 3D-view of brain MRI. Data augmentation techniques are employed to address the challenges of overfitting. The study compares the performance of the proposed models with and without data augmentation for multi-class AD classification, highlighting the significance of augmentation in improving testing accuracy and preventing overfitting.

Huang, Yang & Lee (2021) introduces Federated Conditional Mutual Learning (FedCM) as a novel framework to enhance the performance of FL in Alzheimer’s disease classification. The methodology considers clients’ local performance and the similarity between clients to address the data domain shift issues in MRI data. This is achieved by introducing conditional mutual learning, improving model performance and privacy preservation simultaneously. The use of 3DCNN for T1w MRI classification represents a distinctive contribution to the literature. Khalil et al. (2023) uses the methodology which involves the utilization of FL to train a shared model without sharing raw data with a central server, ensuring privacy. Blood biosample datasets from the ADNI website are employed for model development and evaluation. A distinctive contribution is the introduction of hardware acceleration to expedite training and testing procedures. Blood biosamples are presented as a non-invasive and cost-effective means for diagnosis. The proposed FL and hardware-accelerated approach are introduced as a solution to existing challenges, offering improved accuracy, reduced training time, and lower resource requirements.

Poirot et al. (2019) propose a SL-based approach, a type of distributed learning, and apply it for the first time in the medical field. The methodology involves comparing the performance of SL configurations against centrally hosted and non-collaborative setups. Two medical deep learning tasks are used: a binary classification problem with a dataset of 9,000 fundus photos and a multi-label classification problem with a dataset of 156,535 chest X-rays. The challenges of small sample sizes, privacy concerns, and the burden of multi-center studies are highlighted as limitations that the proposed SL aims to address.

Vepakomma et al. (2018) extends the split neural networks (SplitNN) framework to propose configurations catering to practical health settings, addressing challenges in collaborative training for entities with different modalities of patient data. It explores scenarios where centralized and local health entities collaborate on various tasks and where learning occurs without sharing labels. The study compares SplitNN with other distributed deep learning methods, including FL and large batch synchronous stochastic gradient descent. Results on CIFAR 10 and CIFAR 100 datasets using VGG and ResNet-50 architectures demonstrate that SplitNN achieves higher accuracies with significantly lower computational requirements on the client side compared to alternative methods.

Joshi et al. (2021) explores the efficiency of multi-head split learning (MHSL) as a means to enhance SplitFed Learning (SFL) by eliminating client-side model synchronization. Using a ResNet-18 model on MNIST data, MHSL is studied without synchronization, comparing it to SplitFedV2 on MNIST and CIFAR-10 datasets with five clients. The centralized training of ResNet-18, serving as a baseline, keeps the entire model on the server without splitting, allowing access to all data. Results indicate MHSL’s feasibility, suggesting the potential removal of synchronization in SFL for reduced communication and computation overhead. Experiments with different model portion sizes show negligible impact on overall performance, with SFL offering only a marginal 1 percentage to 2 percentage accuracy improvement over MHSL on the MNIST test set.

Creswell et al. (2018) surveys GANs, covering background, theory, and implementation models. It highlights challenges like mode collapse and training instability, discussing solutions such as modifying distance measures and addressing saddle points. The review also points out open-ended questions in GAN research, including the existence of equilibrium and challenges in evaluating generative models.

Jafarigol & Trafalis (2024) employs data augmentation techniques, including Synthetic Minority Over-sampling TEchnique (SMOTE) and various GANs models, for addressing imbalanced datasets. In the federated setting, CGANs and WGANs-GP outperform other models, emphasizing the effectiveness of synthetic sample generation in mitigating imbalanced weather datasets. The study concludes that GANs-based augmentation, particularly CGANs and Wasserstein GAN + Gradient Penalty (WGANs-GP, enhances model accuracy in both centralized and federated settings. Dong et al. (2023) introduces Federated Semi-supervised Learning for Class Variable Imbalance (FCVI) to address class-variable imbalance. FCVI utilizes a class-variable learning algorithm within a federated setting, consistently outperforming baseline methods like K-SMOTE, and federated averaging (FedAvg) in scenarios involving changes in class numbers and distributions across different training rounds.

Ullah et al. (2023) explores the use of federated learning (FL) for accurate brain tumor segmentation while preserving patient data privacy. The proposed framework leverages a U-Net-based model known for its exceptional performance in semantic segmentation tasks. FL enables collaborative learning by training the segmentation model on distributed data from multiple medical institutions without sharing raw data. The scalability of FL is emphasized, showcasing improved specificity to 0.96 and a dice coefficient of 0.89 with an increase in clients from 50 to 100.

In Thapa et al. (2022), the researchers implemented SFL across CIFAR-10, MNIST, FMNIST, and HAM10000 datasets, employing diverse deep learning models including ResNet18, AlexNet, and LeNet. Two distinct algorithms, namely SFL v1 (with label sharing) and SFL v2 (without label sharing), were introduced. The study conducted a comprehensive comparison with FL, SL and the two SFL variations. The empirical findings demonstrate that SFL achieves comparable test accuracy and communication efficiency to SL while significantly reducing computation time per global epoch in scenarios with multiple clients.

Table 1 shows the comparison of the related works with the dataset used and the results and interpretations.

Table 1 Comparison of related works.

Method	Dataset	Result and interpretation	
FedCM (Federated Conditional Mutual Learning) (Huang, Yang & Lee, 2021)	Alzheimer	In the 3-class classification, the method achieved accuracy rates of 74.5%, 76.0%, and 76.0% on the OASIS, AIBL-1, and AIBL-2 datasets respectively. Also, the FedCM approach improved the unweighted accuracy in the 2-class classification by 6.8%, 2.4%, and −0.6%.	
SFL v1 (with label sharing) and v2 (without label sharing) (Thapa et al., 2022)	HAM10000	The article presents two models applied to the dataset: ResNet and AlexNet. In Version 1 (V1), test accuracies of 79% and 70.5% were achieved for ResNet and AlexNet respectively. In Version 2 (V2), improved test accuracies of 79.2% and 74.9% were attained for the same models.	
FL, SL (Thapa et al., 2022)	HAM10000	The article introduces two models, ResNet and AlexNet, which were both used on the dataset. In FL, these models achieved test accuracies of 77.5% and 75% respectively.In SL, the test accuracies of 79.1% and 73.8% were attained for the respective models.	
FL (Roth et al., 2020)	Mammography breast density classification dataset	The authors achieved an average performance of local models of 0.68 in the FL setting, confirming the ability of FL to achieve models comparable to models trained when the data is accumulated in a central database.	
Multi Head SL (Joshi et al., 2022)	HAM10000, ECG	In SFL, client-side model synchronization adds communication and computation burdens, especially in resource-limited environments. To boost learning efficiency, the article suggests SFL without this synchronization, introducing MHSL. Test accuracies for datasets are 79.56% and 71.04%, respectively.	
Federated Learning with Shared Label Distribution (FedSLD) (Luo & Wu, 2022)	OrganMNIST(axial) and the PathMNIST	In practical non-IID settings, the Best Mean Client Test Accuracy (BMCTA) and Best Test Accuracy (BTA) achieved for the organMNIST dataset are 84.75% and 84.75% respectively. For the PathMNIST dataset, the corresponding values are 53.87% and 57.90% respectively.	
SFL (Joshi et al., 2022)	HAM10000, ECG	The results find that SFL provides 81.37% and 73.40% better accuracy than MHSL on the ECG and HAM-10000 datasets	
FL with GAN (Roy, Rahman & Ahmed, 2022)	Alzheimer	Algorithm used in FL are VGG16 and Xception, both have achieved an accuracy of 97.8 percentage and 93.1 perecentage respectively.	

Methods

The proposed system leverages SFL to reconcile the privacy benefits of SL and the efficiency of FL. SFL addresses the drawbacks of FL and SL by offering model privacy through network splitting and differentially private client-side model updates, while also enabling parallel processing across clients to improve training speed. The study aims to develop a GAN-based image augmentation system and implement an SFL model for Alzheimer’s disease detection, ensuring patient privacy. The objective is to detect Alzheimer’s disease from distributed datasets, training them in a decentralized manner across multiple client servers, with each client representing an individual hospital. Figure 1 depicts the flow of the proposed system. Every client is an individual hospital in Fig. 1.

Figure 1 Flowchart of split federated learning with conditional GAN and image enhancement for alzheimer’s disease detection.

The specific objectives of this research include:

1. Addressing data imbalance in clients through the implementation of cGANs.

2. Using SFL to preserve patient privacy and comply with data protection regulations.

3. Achieving Alzheimer’s disease detection with a satisfactory level of accuracy using ResNet18 model architecture.

The system employs SFL by combining the parallel processing capabilities of FL and model splitting from SL. SL enables training on clients with limited computing resources by focusing on the initial layers of the split ML network model. The proposed system consists of three integral components as depicted in Fig. 2: the client side, split server, and fed server. Clients use GANs to address data imbalance, and the SFL model is applied to an Alzheimer’s disease detection dataset, executed in Google Colab Pro with high-RAM settings. In this distributed approach, the model is partitioned between decentralized clients and a split server. Clients use cGAN to balance data, initiate local training, and calculate gradients (smashed data). The smashed data is sent to the split server, which manages model building, processes back-propagation, and sends gradients back to clients. The fed server uses federated averaging to consolidate updates from diverse clients and sends back updated gradients for further enhancement. This iterative process continues until satisfactory accuracy is achieved.

Figure 2 Split federated learning with cGAN.

Split federated learning with conditional generative adversarial network.

Data collection and preprocessing

The dataset, sourced from Kaggle, consists of MRI scan images associated with Alzheimer’s disease, categorized into four classes: “Very Mild Demented”, “Mild Demented”, “Moderate Demented”, and “Non Demented”, with corresponding image counts of 2,240, 896, 64, and 3,200, respectively. In this study, we will refer to these classes as “Very Mild Dementia”, “Mild Dementia”, “Moderate Dementia”, and “Non-Dementia”. However, the Kaggle dataset used in our research does not include comprehensive demographic information such as gender distribution, age, or dementia onset age for the participants. This limitation has been noted to ensure transparency and inform readers of the constraints faced in this study. The dataset only specifies the number of subjects in each dementia class, which are as follows: Mild Dementia Class: 28 subjects, Moderate Dementia Class: two subjects, Non Dementia Class: 100 subjects and Very Mild Dementia Class: 70 subjects. All the four classes have varied sample distribution, this difference in sample size is known as data imbalance. For this research, only three classes will be considered: “Mild Dementia”, “Moderate Dementia”, and “Non Dementia”. Additionally, only a quarter (1/4th) of the dataset will be utilized while preserving the imbalance ratio of the dataset. The decision to use only three classes and include just a quarter of the dataset was driven by practical constraints. Due to the large size of the dataset, it was necessary to reduce the number of images to facilitate initial trial runs and debugging processes. Even with access to the paid version of Google Colab Pro, technical dependencies required managing a smaller subset of images for effective code testing and troubleshooting. Imbalance ratio formula is given in (1), the value scales from 0 to 1. They are indirectly proportional, lower the value higher is the imbalance of the dataset and vice versa. Imbalance ratio of “Mild Dementia” (minority class 1) to “Non Dementia” is 0.28. Imbalance ratio of “Moderate Dementia” (minority class 2) to “Non Dementia” is 0.02. (1) Imbalance Ratio=Number of Minority Class SamplesNumber of Majority Class Samples.

Figure 3 illustrates a subset comprising one-fourth of the dataset samples, specifically on the chosen three classes. The preprocessing of the dataset is written in Python and implemented using PyTorch to prepare datasets and data loaders for Alzheimer’s disease detection task. Extensive data augmentation is applied including random resizing, flipping, rotation, color jitter, and more, to the training dataset for improved model generalization. The test dataset is transformed with resizing and normalization. The DataLoader iterators are configured with a batch size of 64, and the training loader shuffles data for each epoch. The code is designed to enhance the robustness of a neural network model by introducing varied transformations to the training data. The dataset is partitioned into 80 percentage for training and 20 percentage for testing.

Figure 3 Alzheimer’s dataset description.

Classes vs No. of samples in Alzheimer’s dataset.

Implementation of generative adversarial networks

GANs are a class of machine learning models that consist of a generator and a discriminator trained simultaneously through adversarial training. The generator generates synthetic data, and the discriminator distinguishes between real and synthetic samples. (2) ExlogDx+Ezlog1−DGz.

The generator aims to minimize the (2) function, while the discriminator strives to maximize it. D(x) denotes the discriminator’s evaluation of the probability that a real data instance (x) is indeed real. Ex signifies the expected value over all real data instances. G(z) represents the output of the generator when supplied with random noise (z). D(G(z)) stands for the discriminator’s probability estimate that a generated instance is real. Ez represents the expected value over all random inputs to the generator, effectively covering all generated fake instances G(z). Equation (2) is the cross-entropy between the real and generated distributions. The log(D(x)) term is beyond the direct influence of the generator. Consequently, for the generator, minimizing the loss is analogous to minimizing log (1 - D(G(z))).

In the context of imbalanced medical datasets, GANs can be employed to generate synthetic samples for the minority classes, effectively addressing the data imbalance issue. A cGAN is a type of GAN model. It generates new images, based on specific conditions provided during training. This allows for more controlled and targeted generation of content compared to normal GANs. Among the three classes, two classes, namely “Mild Dementia” and “Moderate Dementia”, have a lower number of samples, classifying them as part of the minority class category. The class labeled “Non Dementia” is referred to as the majority class. cGAN is trained on the dataset to provide real synthetic images for the minority classes. The cGAN generator takes random noise, conditions also known as labels and image as input. It aims to produce synthetic data that resemble the target distribution class. Table 2 illustrates the layers, output shape and parameters of the generator model. The cGAN discriminator evaluates the authenticity of input samples by distinguishing between real and generated data. Table 3 shows the layers, output shape and parameters of the discriminator model. Algorithm 1 depicts the implementation of cGAN.

Algorithm 1:	
Data preprocessing:	
1. Load images from the dataset.	
2. Resize each image to a target size (256 × 256).	
3. Normalize image pixel values to the range [0,1].	
4. Map class labels to integers.	
Define discriminator model:	
Define a function ‘define_discriminator(in_shape =(256, 256, 1))’ to:	
(a) Create a label input and embed it.	
(b) Reshape the embedding to match the image dimensions.	
(c) Concatenate the image input with the label embedding.	
(d) Add multiple Conv2D layers with LeakyReLU activation.	
(e) Flatten the output and add a Dropout layer.	
(f) Add a Dense layer with sigmoid activation for binary classification.	
(g) Compile the model with ‘Adam’ optimizer and ‘binary_crossentropy’ loss.	
Define generator model:	
Define a function ‘define_generator(latent_dim, n_classes =3)’ to:	
(a) Create a label input and embed it.	
(b) Reshape the embedding to match the latent dimension.	
(c) Create a latent input and reshape it.	
(d) Concatenate the latent input with the label embedding.	
(e) Add UpSampling2D and Conv2DTranspose layers to upsample to the target image size.	
(f) Use LeakyReLU activations and a final Conv2DTranspose layer with ‘tanh’ activation.	
Define GAN model:	
Define a function ‘define_gan(g_model, d_model)’ to:	
(a) Set the discriminator as non-trainable.	
(b) Combine the generator and discriminator models.	
(c) Compile the combined model with ‘Adam’ optimizer and ‘binary_crossentropy’ loss.	
Training preparation:	
(a) ‘load_real_samples(images, labels)’ to convert lists of images and labels to numpy arrays.	
(b) ‘generate_real_samples(dataset, n_samples)’ to randomly select real samples from the dataset.	
(c) ‘generate_latent_points(latent_dim, n_samples, n_classes)’ to generate random points in latent space with corresponding labels.	
(d) ‘generate_fake_samples(generator, latent_dim, n_samples)’ to generate fake samples using the generator.	
Train the GAN:	
Define a function ‘train(g_model, d_model, gan_model, dataset, latent_dim, n_epochs =100, n_batch =128)’ to:	
(a) Iterate over epochs.	
(b) For each epoch, iterate over batches:	
      (i) Generate real and fake samples.	
      (ii) Train the discriminator on real and fake samples separately.	
      (iii) Prepare latent points and labels for the generator.	
      (iv) Train the generator via the combined GAN model.	
(c) Save the generator model periodically.	
Load the trained generator model:	
(a) Generate images for a specific class using latent points and labels.	
(b) Save generated images to the specified Google Drive folder.	

Both generator and discriminator are trained simultaneously through adversarial learning for 1,000 epochs achieving average accuracy on real samples and fake samples such as 96.95 percentage and 98.67 percentage, respectively. The average accuracy of real and fake samples refers to the average classification accuracy achieved by the discriminator on real and generated (fake) samples during each training epoch. The discriminator’s loss on real and fake samples stands at 0.117 and 0.048, respectively. Figure 4 displays the generated synthetic images of class Moderate Dementia.

Table 2 Model summary of generator.

Layer	Output shape	Parameters	
Input layer	[(None, 100)]	0	
Input layer	[(None, 1)]	0	
Dense	(None, 4096)	413,696	
Embedding	(None, 1, 50)	200	
LeakyReLU	(None, 4096)	0	
Dense	(None, 1, 4096)	208,896	
Reshape	(None, 64, 64, 1)	0	
Reshape	(None, 64, 64, 1)	0	
Concatenate	(None, 64, 64, 2)	0	
UpSampling2D	(None, 128, 128, 2)	0	
Conv2DTranspose	(None, 256, 256, 64)	2,112	
LeakyReLU	(None, 256, 256, 64)	0	
Conv2DTranspose	(None, 256, 256, 1)	1,025	

Table 3 Model summary of discriminator.

Layer	Output shape	Parameters	
Input layer	[(None, 1)]	0	
Embedding	(None, 1, 50)	200	
Dense	(None, 1, 65,536)	3,342,336	
Input layer	[(256, 256, 1)]	0	
Reshape	(256, 256, 1)	0	
Concatenate	(256, 256, 2)	0	
Conv2D	(None, 128, 128, 128)	2,432	
LeakyReLU	(None, 128, 128, 128)	0	
Conv2D	(None, 64, 64, 128)	147,584	
LeakyReLU	(None, 64, 64, 128)	0	
Conv2D	(None, 32, 32, 128)	147,584	
LeakyReLU	(None, 32, 32, 128)	0	
Conv2D	(None, 16, 16, 128)	147,584	
LeakyReLU	(None, 16, 16, 128)	0	
Conv2D	(None, 8, 8, 128)	147,584	
LeakyReLU	(None, 8, 8, 128)	0	
Flatten	(None, 8,192)	0	
Dropout	(None, 8,192)	0	
Dense	(None, 1)	8,193	

Figure 4 Synthetic images generated from cGAN.

Image source: https://www.kaggle.com/datasets/yasserhessein/dataset-alzheimer. License: Open Database License (ODbL) v1.0.

Image preprocessing

Image enhancement techniques are performed such as Contrast Limited Adaptive Histogram Equalization (CLAHE) and image resizing on the generated GAN images. CLAHE improves local contrast in the grayscale images and then images are resized to a specified dimension (224 × 224), ensuring a consistent size for further analysis or processing. Figure 5 displays both images with and without enhancement techniques. The generated GAN images are subjected to various transformations. The generated GAN images undergo several preprocessing (data augmentation) steps before applying SFL, including random resized cropping, random horizontal and vertical flipping, random rotation, color jittering, random grayscale conversion, and normalization.

Figure 5 Synthetic images with and without image enhancement.

Image source: https://www.kaggle.com/datasets/yasserhessein/dataset-alzheimer. License: Open Database License (ODbL) v1.0.

Integration of SFL

After balancing the data in the client side, the SFL process is initiated. The ResNet18 algorithm is employed in this study, where the model is partitioned between the client and the split server. ResNet18 is a well-established architecture optimized over years, whereas SFL is relatively new and still evolving. ResNet18 benefits from centralized training with consistent data availability, enhancing its feature learning capabilities. SFL, with its distributed nature, faces challenges in synchronization and data variability across nodes. ResNet18 uses optimized training techniques, while SFL’s federated learning introduces communication overhead and potential synchronization issues, impacting performance. While ResNet18 currently shows better results, SFL’s potential for privacy-preserving distributed learning is significant. Future advancements in SFL optimization and data handling could narrow the performance gap.

The client and server architecture has been implemented in Google Colab as separate functions. The interactions between the server and client are managed through these functions, with the server and client responses being sent back as return values from their respective functions. This setup allows for clear and modular communication within the Colab environment. In Fig. 6, the client-side model follows a simplified ResNet18 architecture, designed for initial feature extraction and dimensionality reduction. It comprises two sequential layers. The first layer incorporates a convolutional operation, batch normalization, rectified linear unit (ReLU) activation, and max pooling to reduce spatial dimensions. The second layer refines features through additional convolutional operations and batch normalization. A dropout layer is applied for regularization. In step 1, the model aims to capture essential features from the input data before transmitting the processed information to the server-side for more complex computations and decision-making. The algorithm for SFL with label sharing (Thapa et al., 2022) is depicted in: Algorithm 2 .

Algorithm 2:	
Initialization:	
1. Define Models:	
          - Client-side ResNet18 model (‘net_glob_client’).	
          - Server-side ResNet18 model (‘net_glob_server’).	
2. Define Functions:	
          - Federated Averaging function (‘FedAvg’).	
          - Server-side training function (‘train_server’).	
          - Server-side evaluation function (‘evaluate_server’).	
3. Initialize Global Variables:	
          Track training and testing metrics.	
Split and Federated Learning Loop:	
For each global training round from 1 to ‘epochs’:	
1. Client-Side Implementation:	
          (a) Random Client Selection:	
                    - Randomly select a fraction ‘frac’ of clients (‘m’) to participate in the current round.	
          (b) Initialize Local Weights Storage:	
                    - Initialize an empty list ‘w_locals_client’ to store local model weights from clients.	
          (c) Local Training:	
                   For each selected client ‘idx’:	
                              i. Instantiate Client:	
                                        - Instantiate a client with the client-side ResNet18 model (‘local’).	
                              ii. Train Local Model:	
                                        - Train the local model using the ‘train’ method with a deep copy of the global client model.	
                                        - Perform a forward pass to obtain smashed data (‘smashed_data’).	
                               iii. Transmit Smashed Data:	
                                        - Transmit the ‘smashed_data’ and true labels (‘y’) to the server.	
2. Server-Side Implementation:	
          (d) Receive Data:	
          - Receive smashed data and true labels from clients.	
          (e) Server-Side Training:	
                              i. Forward Pass:	
                                        - Perform a forward pass on the server-side ResNet18 model using ‘smashed_data’ to obtain predictions (‘ypred’).	
                              ii. Calculate Loss:	
                                        - Calculate loss using ‘y’ and ‘ypred’.	
                              iii. Backward Pass:	
                                        - Compute gradients of the loss with respect to ‘smashed_data’.	
                                        - Send these gradients back to the respective clients.	
3. Client-Side Backpropagation:	
          (f) Update Local Models:	
                                        - Clients receive gradients from the server.	
                                        - Perform backpropagation using the received gradients.	
                                        - Update their local models.	
          (g) Send Local Weights:	
                                        - Clients send their updated model weights to the server.	
4. Server-Side Aggregation:	
          (h) Federated Averaging:	
                           - Perform Federated Averaging (‘FedAvg’) on the local model weights to obtain the global model weights (‘w_glob_client’).	
           (i) Update Global Model:	
                           - Update the global client model (‘net_glob_client’) with the calculated ‘w_glob_client’.	
End	

In Fig. 6, the server-side model, following the ResNet18 architecture, is used for intricate feature extraction and decision-making. In Layer 3, a convolutional operation with batch normalization and ReLU activation is applied, followed by another convolutional operation with batch normalization. Layers 4, 5, and 6 each consist of a Baseblock, which includes two sets of convolutional operations with batch normalization and a shortcut connection to aid learning. This progressive deepening of the network captures features and relationships within the input data. The model concludes with an average pooling layer to reduce spatial dimensions and a fully connected layer for the final classification, yielding three output classes. This hierarchical structure enables the server-side model to discern complex patterns, making it adaptive for decision and classification tasks. In step 2, the server side computes the gradients and sends back to the client through back propagation. The clients update their model after receiving the gradients from the server. In Fig. 6, each client transmits its weights to the federated server in step 3, which subsequently employs the Federated Averaging algorithm. During step 4, the server returns the averaged weight to each client, facilitating the update of their respective models.

Figure 6 ResNet18 architecture in split federated learning.

Privacy considerations in this SFL framework are addressed through several key mechanisms. Firstly, data locality ensures that clients retain their raw data locally, transmitting only intermediate representations, known as smashed data, to the server. This approach significantly reduces the risk of sensitive data exposure. Additionally, gradient encryption can be employed when sending gradients back to clients, further safeguarding the privacy of the data. The split model design inherently reduces data exposure by ensuring that only part of the computation is handled by the server, thereby minimizing the risk of data leakage and enhancing the overall privacy of the FL process.

Results and Discussion

Table 4 outlines the key parameters and configurations used in the training process of FL, SL and SFL.

Table 4 Training configuration of proposed model architecture.

Optimizer	Adam	
Batch size	64	
Learning rate	0.0001	
Epochs	50	
No. of classes	3	
No. of Clients	2	
Model	ResNet18	
Loss	Categorical cross entropy	

The optimizer employed is Adam. A batch size of 64 indicates the number of data samples used in each iteration to update the model parameters, which can affect convergence and efficiency. The learning rate, set at 0.0001, determines the step size in the parameter space during optimization, playing a crucial role in balancing convergence speed and stability. The training is conducted over 50 epochs, representing the number of complete passes through the entire dataset, allowing the model to be refined over successive iterations. With three classes, the model is designed to extract and learn relevant representations from the input data. Categorical cross-entropy is used as the loss function for multi-class classification tasks, enabling the quantification of the model’s performance.

The result is obtained after training the SFL model on the Alzheimer’s dataset with two clients for 50 epochs. It shows result for two models (i) case 1 without GAN images, (ii) case 2 with GAN images. Generating too many instances for the minority class may lead to overfitting, where the model becomes too specific to the synthetic data and fails to generalize well on real-world data. Minimizing the majority class too much could lead to the model becoming biased towards the minority classes, impacting its performance on the majority class. GANs can only generate data that resembles the real data they were trained on. There is a limit to how much they can diversify the minority class data without compromising its validity. In some cases, maintaining a certain level of imbalance may be necessary to reflect the real-world distribution of the data accurately. Figures 7A and 7B represents the distribution of image samples before and after the inclusion of real synthetic images into the dataset respectively. The pie charts in the Fig. 7 details the count and percentage of samples for all three classes.

Figure 7 Image sample distribution (A) Case 1–without GAN and (B) Case 2–with GAN.

Table 5 provides a comparative analysis between SFL models without and with GANs. In the SFL without GAN, the training accuracy is slightly higher at 81.71% compared to 78.40% with GANs. However, the model with GANs achieves a lower training loss (0.38 vs. 0.48). The testing accuracy is also higher with GANs (80.48% vs. 83.54%), the testing loss is significantly lower with GANs (0.38 vs. 0.51). Furthermore, when considering precision and recall which are the essential metrics in medical imaging, the model with GANs outperforms significantly. With GANs, the precision reaches a perfect score of 1.00, highlighting the model’s ability to minimize false positives, crucial in medical diagnoses. Additionally, the recall value substantially improves with GANs (0.84 vs. 0.53), indicating the model’s effectiveness in capturing relevant instances within the dataset, thereby reducing false negatives. The use of GANs typically increases the size and diversity of the training dataset, necessitating more epochs for the model to fully learn from the larger dataset. Models trained with GANs often exhibit different learning dynamics, with additional images introducing more variability and requiring longer training to achieve convergence. However, after a certain point, the model may start overfitting or experiencing diminishing returns, leading to decreased performance. The model with GAN achieves these superior results slightly later, at the 40th epoch compared to the 30th epoch without GANs. For the model without GAN, convergence was observed by epoch 30, leading to stop the training to save time and resources. Empirical observations from preliminary experiments showed that extending training beyond 30 epochs did not yield significant improvements. Validation performance indicated that the model had reached optimal performance by epoch 30, making further training unnecessary.

Table 5 SFL metrics.

Metric	Case 1–Without GAN	Case 2–With GAN	
Train accuracy	81.71	78.40	
Train loss	0.38	0.48	
Test accuracy	80.48	83.54	
Test loss	0.51	0.38	
Precision	0.23	1.00	
Recall	0.53	0.84	
Epoch at which result achieved	30	40	

The case 2 model is trained along with the synthetically generated images of the minority classes and it will be able to classify Alzheimer’s disease detection with 83.54 percentage test accuracy. This result shows the impact of addressing data imbalance in SFL across diverse clients, illustrating how such imbalances can adversely affect model training. The accuracy is based on multiple runs of the code. Specifically, two to three independent tests were conducted where the data was split into training and test sets, and the results remained consistent across all iterations, showing minimal variance in the reported accuracy. The consistency of the outcomes across these runs further strengthens the validity of the reported results. In conclusion, the addition of GAN images enhances the model’s performance in medical image analysis. These metrics such as testing loss, precision, and recall values show the importance of GANs in augmenting datasets and improving the robustness and accuracy of machine learning models, particularly in medical applications where precise diagnoses and minimal errors are crucial. The graphs in Fig. 8 illustrates the training and testing performance for case 2 up to the 40th epoch. The training accuracy in Fig. 8A demonstrates a consistent improvement, reaching a peak of 78%. The testing accuracy Fig. 8B, although fluctuating between 60% and 80% during training, eventually stabilizes and achieves a final value of 83% percentage. Figure 9 displays the train and test loss plotted against the number of epochs. The training loss in Fig. 9A consistently decreases with increasing epochs. Meanwhile, the testing loss in Fig. 9B fluctuates heavily but finally reaches 0.47 at the end of the 40th epoch. Beyond the 40th epoch, the training accuracy continues to rise; however, the testing accuracy experiences a sharp decline, indicating potential overfitting of the model. The accuracy presented in the manuscript is based on multiple runs. Specifically, two to three independent tests were conducted, where the data was split into training and test sets. The results were consistent across all iterations, showing minimal variance in the reported accuracy. The consistency of the outcomes across these runs further strengthens the validity of the reported results.

Figure 8 Plot results for (A) training and (B) testing accuracy.

Figure 9 Plot results for train and test loss.

Table 6 indicates the individual train accuracy, train loss, test accuracy, test loss, precision and recall for both the clients.

Table 7 presents a comparison of performance metrics accuracy, loss, precision, and recall across three different machine learning methodologies after adding the GAN images to address the data imbalance: FL, SL and SFL. In FL, the accuracy stands at 82.46%, with a corresponding loss of 0.52. All three learning methods achieve precision value as 1.00, indicating no false positives, they differ in their recall rates. SL has the highest recall at 0.93, followed by SFL at 0.84, and FL at 0.83. This suggests that SL is the most sensitive to true positives among the three methods. In SL, the accuracy slightly decreases to 80.13%, accompanied by a lower loss of 0.45. However, precision remains 1, while recall improves to 0.93, implying a reduction in false negatives. SFL exhibits the highest accuracy among the three approaches, reaching 83.54%, with the lowest loss of 0.38. Precision remains 1.00, and recall is maintained at 0.84, indicating a good balance between true positive rate and false negative rate. Overall, SFL outperforms both FL and SL in terms of accuracy and loss metrics, while maintaining perfect precision. Although SL shows a higher recall than FL, SFL achieves a comparable recall while also offering superior accuracy and loss values.

Table 6 Individual performance metrics of clients in SFL with GAN.

SFL–With GAN	Client 1	Client 2	
Train accuracy	77.08	79.71	
Train loss	0.50	0.46	
Test accuracy	84.34	82.75	
Test loss	0.38	0.38	
Precision	1.00	1.00	
Recall	0.89	0.80	

Table 7 Performance of FL, SL and SFL with GAN images indicates the individual train accuracy, train loss, test accuracy, test loss, precision and recall for both the clients.

Metrics	FL	SL	SFL	
Accuracy	82.46	80.13	83.54	
Loss	0.52	0.45	0.38	
Precision	1.00	1.00	1.00	
Recall	0.83	0.93	0.84	

Table 8 presents the execution times for different learning methodologies: FL requires the longest execution time of 40 s, likely due to the decentralized nature of the training process, which involves communication between multiple devices or clients. SL reduces the execution time to 35 s, indicating improved efficiency compared to FL. This reduction may be from the partitioning of the model into two parts: one on the client device and one on the central server, allowing for parallel computation. Interestingly, both SFL and SL show the same execution time of 35 s. SFL combines elements of both FL and SL methodologies. It partitions the model for parallel computation while also leveraging the decentralized training approach of FL. This combination aims to achieve efficient learning without compromising on privacy or data locality.

Table 8 Computational time for FL, SL and SFL.

Methods	Execution time	
FL	40s	
SL	35s	
SFL	35s	

Table 9 outlines the performance metrics of various SFL methods applied to different medical datasets. Each row represents a different SFL approach along with the corresponding dataset, train accuracy, and test accuracy: SFL v1 with label sharing utilizing ResNet achieved a train accuracy of 90.2% and a test accuracy of 79% on the HAM1000 dataset. Similarly, SFL v1 with label sharing using AlexNet attained a train accuracy of 69% and a test accuracy of 70.5% on the same dataset. SFL v2 without label sharing employing ResNet achieved outstanding train accuracy of 99.3% and a test accuracy of 79.2% on the HAM10000 dataset. On the same dataset, SFL v2 without label sharing using AlexNet obtained a train accuracy of 72.1% and a test accuracy of 74.9%. In Dong et al. (2023), the SFL method on HAM10000 dataset achieved a train accuracy of 78.8% and a test accuracy of 81.37%. On the ECG dataset, it yielded a train accuracy of 80.98% and a test accuracy of 73.40%.

Table 9 Analysis of the proposed method in comparison to other approaches.

Methods	Dataset	Train accuracy	Test accuracy	
SFL v1 with label sharing using ResNet (Thapa et al., 2022)	HAM10000	90.2	79	
SFL v1 with label sharing using AlexNet (Thapa et al., 2022)	HAM10000	69	70.5	
SFL v2 without label sharing using ResNet (Thapa et al., 2022)	HAM10000	99.3	79.2	
SFL v2 without label sharing using AlexNet (Thapa et al., 2022)	HAM10000	72.1	74.9	
SFL (Joshi et al., 2022)	HAM10000	78.8	81.37	
SFL (Joshi et al., 2022)	ECG	80.98	73.40	
Proposed–SFL with GAN	Alzheimer	78.4	83.54	

Conclusions

The current study successfully addresses critical challenges in the healthcare sector by developing an innovative medical classification model for Alzheimer’s disease detection. One of the key challenges in medical data analysis is data imbalance, where certain classes are unevenly represented, leading to potential bias in machine learning models. The SFL method, particularly suitable for medical data analysis, uses network splitting to accommodate low-computing resources and parallel handling of clients to ensure fast training, crucial for time-sensitive medical applications. Additionally, SFL enables secure analysis of private and sensitive medical data, preserving patient confidentiality while advancing diagnostic and treatment capabilities. In this study, the integration of SFL and GANs enhances data privacy by enabling collaborative model training without sharing raw data. GANs contribute by generating realistic synthetic samples to mitigate imbalanced data challenges in medical classification. The addition of synthetic real images improves test accuracy, demonstrating the impact of addressing data imbalance in SFL across diverse clients.

However, a drawback of this approach is the significant fluctuation in test metrics, making it challenging to determine the optimal stopping point for training. The proposed method, SFL with GAN, applied to the Alzheimer dataset uses ResNet model architecture that has achieved a train accuracy of 78.4% and a test accuracy of 83.54%. Overall, the proposed SFL method outperforms other approaches in terms of test accuracy, demonstrating its effectiveness in medical dataset analysis. Future work will focus on several areas to enhance the current methods. These include developing a multi-class classification model for more than three classes, utilizing non-IID (non-independent and identically distributed) datasets, exploring different splitting strategies, and addressing the issue of fluctuating test metrics. Additionally, applying the current methods to other datasets and scaling up to larger datasets will be crucial for broader applicability and validation of the approach. Incorporating random replications of subsets of the data to better quantify the variance and robustness of the model’s test accuracy will also be a priority. By addressing these limitations and pursuing these future directions, the potential of SFL in advancing medical data analysis and improving healthcare outcomes can be further realized.

Supplemental Information

Supplemental Information 1 Real and synthetic Images for Mild Dementia, Moderate Dementia and Non- Dementia classes for training and testing

Image source: https://www.kaggle.com/datasets/tourist55/alzheimers-dataset-4-class-of-images. License: Open Database License (ODbL) v1.0.

Supplemental Information 2 Real Images for Mild Dementia, Moderate Dementia and Non-Dementia classes for training and testing

Image source: https://www.kaggle.com/datasets/tourist55/alzheimers-dataset-4-class-of-images. License: Open Database License (ODbL) v1.0.

I extend my heartfelt gratitude to everyone who has contributed to this endeavor, including my mentors, colleagues, and friends. Their insights, feedback, and collaboration have enriched the research and enhanced its impact. I also acknowledge the resources and facilities provided by my college Vellore Institute of Technology (VIT), which have been essential in conducting the research effectively. In conclusion, I am truly grateful for the opportunity to undertake this research project, and I look forward to sharing its findings with the academic community.

Additional Information and Declarations

Competing Interests

Author Contributions

Data Availability

The authors declare there are no competing interests.

G Narayanee Nimeshika performed the experiments, performed the computation work, prepared figures and/or tables, and approved the final draft.

Subitha D conceived and designed the experiments, analyzed the data, authored or reviewed drafts of the article, and approved the final draft.

The following information was supplied regarding data availability:

The code is available at Zenodo: Narayanee Nimeshika. (2024). Nimeshika5/Split-Federated-Learning-for-Alzheimer-s-disease-CODE: Initial Release (0.1.0). Zenodo. https://doi.org/10.5281/zenodo.10899890.

The raw data is available at Zenodo: Narayanee Nimeshika. (2024). Nimeshika5/Alzheimer-Dataset: Final dataset (1.1.0). Zenodo. https://doi.org/10.5281/zenodo.10948116.

The third-party data is available at Kaggle:

https://www.kaggle.com/datasets/yasserhessein/dataset-alzheimer.

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
