# Peer review of "Enhancing Alzheimer’s disease classification through split federated learning and GANs for imbalanced datasets"

_PeerJ Computer Science, doi:10.7717/peerj-cs.2459_

## Round 0.1 · original submission · Major Revisions

The authors need to revise the manuscript as per the reviewer's comments.

Reviewer 1 ·

Basic reporting

The paper proposed an interesting approach by using Split Federate Learning (SFL) to classify different Alzheimer’s disease conditions to address the challenges in applying AI in healthcare. The authors showed that the SFL results ranked the best compared to Federated Learning or Split Learning.
The concept of this SFL is interesting to the field, but there are a few questions the author should address to improve the quality of the paper.
1. The authors used a Kaggle dataset without a detailed description of the participant’s characteristics. I recommend that the authors add more descriptions of the participants involved in the study, such as gender distribution, age, dementia onset age, etc. In this way, the arguments would be stronger for the scientific community.
2. Using GAN to overcome the issue of imbalanced data is good for training the models, but it would be great if the authors could discuss the biological relevance of the imputed data, or that’s a limitation of the study.
3. Regarding the implementation of the SFL, the authors described the concept nicely in the introduction, but the actual implementation seems vague to me. It would be more helpful to know how the authors implemented the SFL with two different parts, the central server and client sites, using Google Colab. Moreover, it would be great if the authors could discuss how, in reality, this approach would protect privacy.
4. No code was included in the submission.
5. The report associated with the Kaggle dataset performs better using ResNet18 than the SFL model reported in the paper. The author should discuss the differences and offer a potential explanation if the ResNet18 learned the same features from different approaches.

Experimental design

The comparisons between SFL and FL/SL are good. The authors should clearly address the current study's limitations and propose the future direction of applying the current methods to other datasets and scaling up to large datasets.

Validity of the findings

The results presented in the current paper split the data into training and testing sets to validate the model performance. It is recommended that the authors deposit the code and provide access to the scientific community.

Additional comments

1. The in-text citation formatting is mixed up. It is recommended that the author use a citation manager to organize the references.
2. The authors need to provide more descriptive Figure legends.
3. The organization of the paper is a bit weird. The authors could move some of the figures/tables to the supplement and keep the main finding in the main figures/tables to make the paper more concise.
4. Please carefully check the grammar and add missing citations to the whole paper.
5. Some parts of the paper seem to be repetitive. Please revise and made the paper more concise.

·

Basic reporting

• The primary goal of this paper is to develop a medical classification model of Alzheimer’s disease classification that can effectively learn from decentralized and imbalanced datasets without compromising on data privacy.
• By incorporating Split Federated Learning (SFL), the paper aims to enable efficient set of distributed agents that collaboratively train learning models without sharing their data, thus improving data privacy and reducing the time/communication overhead.
• The integration of GANs aims to improve the model's ability to generalize across imbalanced classes by generating realistic synthetic samples for minority classes.
• SFL with GAN approach significantly improves model accuracy, recall and precision highlighting the potential for innovative solutions to advance healthcare outcomes.

Experimental design

• SFL is applied on Alzheimer’s disease detection dataset and all the python scripts and models are executed in Google Colab Pro with a high-RAM run-time setting and V100 GPU.
• As a preliminary step, the Alzheimer’s image classification dataset undergoes preprocessing and augmentation techniques.
• Conditional GANs (cGANs) generate synthetic samples for minority classes, addressing data imbalance. Image Enhancement techniques like CLAHE (Contrast Limited Adaptive Histogram Equalization) for improving local contrast in the grayscale images.
• The proposed system adopts Split Federated Learning (SFL), leveraging the parallel processing capabilities of Federated Learning (FL) while incorporating model splitting during training.
• The proposed system comprises three integral components: the client side, split server, and fed server. Clients employ Conditional Generative Adversarial Networks (cGANs) to address data imbalance challenges.

Validity of the findings

• The result presents a comprehensive comparison of performance metrics between different models, demonstrating the impact of incorporating GANs on model performance.
• The result highlights the importance of precision and recall in medical imaging tasks, emphasizing the model's ability to minimize false positives and false negatives.
• The discussion of results includes insights into the epoch at which the desired results are achieved, providing context for model convergence.
• A comparative analysis is provided across various machine learning methodologies such as Federated Learning (FL) and Split Learning (SL), demonstrating the effectiveness of Split Federated Learning with GANs in improving accuracy and addressing data imbalance.
• The execution time analysis adds another dimension to the validity of findings, showcasing the efficiency of the proposed approach compared to other existing methods. It also provides quantitaive anaysis in terms of accuracy and execution time.

Additional comments

• The conclusion provides insights into future directions, including the exploration of multi-class classification, non-IID datasets, and different splitting strategies, indicating pathways for further research.
• The conclusion acknowledges a drawback regarding significant fluctuation in test metrics, providing transparency about potential challenges associated with the proposed approach.
• The paper emphasizes the significance of the research in the healthcare sector, particularly in improving diagnostic and treatment capabilities for Alzheimer's disease while ensuring patient privacy.

Reviewer 3 ·

Basic reporting

The approach followed by the researchers and the goal of the study are interesting and have potential; however, several issues exist in the present version of the analysis and manuscript.
There are some technical issues related to the scientific writing rules.
• Words should be used for numbers from zero through nine, and numerals should be used from 10 onward.
o This can be found in the manuscript as an example in lines 12, 304, 308, etc.
• The citations should be numbered consecutively according to the first mention of each source in the text.
o The first mentioned citation number is [14], as found in line 36.
• When a phrase can be abbreviated, it should be spelled out in full, and the abbreviation should be provided in parentheses at the first mention. After that, you should only use the abbreviation.
o The authors repeat certain phrases with their abbreviations several times, e.g., GAN, SFL, etc.
o However, they listed some abbreviations without the phrase in lines 53, 110, 131, 132, and so on.
• Some paragraphs require references to substantiate the information they present.
o The first paragraph in the introduction spans lines 27 to 35.
o The paragraph starts at line 57.
o Table 1's second row
There are some typos in the writing, which in certain places affect the analysis.
• Line 226: The authors should refer to Table 2, not Table 1.
• Line 228: The authors should refer to Table 3, not Table 2.
• Lines 314–315: The information in Table 5 contradicts the sentence referring to the training loss.
• Lines 343–344: The sentence refers to precision, and recall isn’t consistent with the information presented in Table 7.

In the Related work section
• The authors describe certain studies that relate to their problem; however, in Table 1, they present different studies. Why didn't you summarize all the related work, whereas the studies mentioned in the text are more relevant to the Alzheimer classification problem?

• Figure 3 is uninformative.
• Figure 4: If the figure presents the real image and the generated/fake one, it would be an amazing addition.

Experimental design

Abstract section
• The word “novel” isn’t appropriate here. Just applying two algorithms that work separately isn’t considered a novelty. Feel free to disagree with

Methods section
• It is possible to combine the first two wordy paragraphs into one informative paragraph.
• The authors didn’t mention any technical details about the type of dataset. They just mentioned that it consists of an MRI scan. Could you specify the type of MRI images and the parameters used for their acquisition?
• Additionally, the authors didn’t mention why they chose to use only three classes or why they only included 1/4th of the dataset.
• It would be more appropriate to divide the paragraph that began with line 235 into a section named image preprocessing.

Validity of the findings

Results and discussion section
• In line 304, I didn’t figure out what the word “3 features” refers to.
• In line 308, I didn’t figure out what the word “four” refers to.
• Figure 7: To me, the data is still unbalanced, so why didn't you generate more data for the minority class or at least minimize the majority class?
• Given that the heavy work using GAN didn't significantly improve the result "from my point of view," have you considered trying simple data augmentation methods?
• Table 5: The authors didn’t mention why they stopped the algorithm without GAN at epoch 30, not 40 like the other one, while 10 epochs may be sufficient to improve the result.
• In Line 385, the authors inappropriately compare the computational cost of the first two algorithms, FL and SL, using GAN, with the algorithm named SFL without GAN. They should instead compare SFL with GAN.
• Table 9 presents comparisons between related works, which is good. However, each row in the table shows different experiment that differ by more than one parameter (like the used model, data, or SFL settings aren’t mentioned) from the other, which is an inappropriate comparison. Additionally, the authors didn’t mention any studies that tackled the same problem, “Alzheimer classification.”
• In conclusion, the discussion didn’t relate to the state-of-the art methods that tackle the same problem.

Conclusion
• The Conclusion does not add much to what is already said in the Results and Discussion sections, and I suggest rewriting the Conclusion to highlight the main findings.

Reviewer 4 ·

Basic reporting

The research presented in this manuscript, namely the enhancement Alzheimer’s disease classification through Split Federated learning and GANs for imbalanced datasets features several timely topic of huge importance in statistical machine learning and data science. The authors present a potential very impactful approach that combines two state-of-the-art methods to address some quintessential challenges in the classification Alzheimer's disease. Handling imbalances in the distribution of labels remains an important topic that continues to receive the attention of several researchers and practitioners, and privacy preserving data mining also remains an active topic of research. The authors deserve credit for proposing a classification approach Alzheimer's disease built on a solid foundation that combines methods under these two paradigms namely Split Federated Learning and Generative Adversarial Networks. If implemented rigorously and effectively this framework has the potential of being very useful!
The paper is well written from the perspective of the authors' great command of the English language.

Experimental design

The setup for computational exploration of the proposal classifier on the chosen data is fairly standard, and perhaps does not herein require any additional comments. Perhaps what is missing here, or at least not made clear enough to the reader, is the fact that the authors do not seem to operate with a computational setup that allows the estimation of the test accuracy on the several random replicates of subsets of the data.

Validity of the findings

Although the proposed framework has the potential of being very useful, very impactful and applicable to this interesting problem of Alzheimer's disease classification, a thorough of the manuscript reveals some deficiencies:

The algorithmic details of split theater learning missing

The algorithmic details of GAN are missing

At the very least, for Computer Science journal of the magnitude of PeerJ Computer Science, some rigorous general details of the key algorithms should be provided

The proposed approach does not develop a framework but instead just applies GAN and Split Federated Learning in Cascade rather than rigorously jointly

While the authors do a great job at presenting previous work and the performances thereof, the game in performance achieved by this work namely 83.2% test accuracy, is not substantial, or at the very least is called underwhelming!

From a purely statistical perspective and in the interest of quantifying aspects of uncertainty in both prediction and error estimation, the computational portion of this manuscript is at best too lean and even borderline limited. In other words, using random replications of the test error so as to have a window into the variance of the estimation would have been ideal.

---

## Round 0.2 · Minor Revisions

Minor revisions must be done for this article.

Reviewer 1 ·

Basic reporting

The author addressed my concerns thoroughly. I recommend to accept this manuscript for publication.

Experimental design

The author addressed my concerns thoroughly. I recommend to accept this manuscript for publication.

Validity of the findings

The author addressed my concerns thoroughly. I recommend to accept this manuscript for publication.

·

Basic reporting

The authors have proposed a medical classification model of Alzheimer’s disease classification that can effectively learn from decentralized and imbalanced datasets without compromising on data privacy.

The authors have also developed set of distributed agents that collaboratively train learning models without sharing their data, thus improving data privacy and reducing the time/communication overhead.

Experimental design

• Split Federated Learning approach in the proposed system , leverages the parallel processing capabilities of Federated Learning (FL) while incorporating model splitting during training.

• To address data inbalance challenges, the proposed system utilizes integral components: the client side, split server, and fed server. Conditional Generative Adversarial Networks (cGANs) is used to address data imbalance challenges.

Validity of the findings

• Results highlights the importance of precision and recall in medical imaging taskd, emphasizing the model's ability to minimize false positives and false negatives.

• A comparative analysis is provided across various machine learning methodologies which demonstrates the effectiveness of SPLmwith GANs to improve acuracy. Time analysis implies the efficiency of the proposed approach when compared to other methods.

Additional comments

The paper emphasizes the significance of the research in the healthcare sector, particularly in improving diagnostic and treatment capabilities for Alzheimer's disease while ensuring patient privacy.

Detailed explanations for algorithm and simulation set up is provided and well discussed in the revised manuscript.

Reviewer 3 ·

Basic reporting

- “Figure 3 has been retained because it gives the reader the idea of the sample distribution and data imbalance among the classes”.

Figure 7 present the same concept.

Experimental design

Methods section
- “The decision to use only three classes and include just a quarter of the dataset was driven by practical constraints. Due to the large size of the dataset, it was necessary to reduce the number of images to facilitate initial trial runs and debugging processes. Even with access to the paid version of Google Colab Pro, technical dependencies required managing a smaller subset of images for effective code testing and troubleshooting”.
Deep learning models, particularly those like ResNet, are indeed prone to overfitting when trained on limited datasets. Ignoring three-quarters of the original dataset could exacerbate this issue, especially if the remaining subset does not adequately represent the diversity of the full dataset. Using the entire dataset, even with fewer augmentations per class, could indeed yield better performance. The authors should reconsider their approach to dataset selection and augmentation strategy, as this could significantly enhance the performance and generalizability of their model. A more balanced approach, leveraging both the full dataset and effective augmentation, could yield better results without overfitting risks.

Validity of the findings

Results and discussion section
- “The generated GAN images undergo Random Resized Cropping, Random Horizontal and Vertical Flipping, Random Rotation, Color Jittering, Random Grayscale Conversion and Normalization before applying SFL”.
Is this mean that you generate new data form generated data! Why you need to do this?
While your study focused on proofing the efficiency of GAN augmentation method.

Reviewer 4 ·

Basic reporting

The authors have addressed most of the comments, remarks, and suggestions I made in my initial review! They have specifically provided a more detailed algorithmic description of their proposed framework, thereby improving the manuscript.

Experimental design

Well done!

Validity of the findings

It could well be that the authors did it, but I do not see my request for a measure of uncertainty around their accuracy! It is surely an improvement over the performance of other authors, but being a single estimate of the accuracy, it lacks the statistically needed inferential power for its validity. In other words, unless we have many replicates of the test errors and then compute some central tendency (average, median, mode) thereof, it's incomplete.

Additional comments

As long as the author can confirm one way or the other that the accuracy reported here is based on more than a single run of their algorithm, I deem this manuscript ready in its present form. In other words, I request that the authors properly and duly indicate how many replications of split the data into training and test sets they performed to compute the report accuracy.

---

## Round 0.3 · accepted · Accept

The paper was very well improved. It can be accepted.